# A Health Status Update of *Myocastor coypus* in Northern Italy

**DOI:** 10.3390/ani14020245

**Published:** 2024-01-12

**Authors:** Arturo Nicoletti, Paola Pregel, Laura Starvaggi Cucuzza, Enrico Bollo, Frine Eleonora Scaglione

**Affiliations:** Department of Veterinary Sciences, University of Turin, 10095 Grugliasco, TO, Italy; arturo.nicoletti1991@gmail.com (A.N.); paola.pregel@unito.it (P.P.); laura.starvaggicucuzza@unito.it (L.S.C.); enrico.bollo@unito.it (E.B.)

**Keywords:** nutria, ecological impact, health status, public health, *Toxoplasma gondii*, zoonoses

## Abstract

**Simple Summary:**

Nutria (*Myocastor coypus*) is a pest animal present in Africa, Europe, North America and Asia that causes agricultural and ecological damages. Moreover, it has to be considered as a potential risk for public health. A health survey on a population of coypu of Northwest Italy has been carried out to evaluate its zoonotic risks. None of the animals were positive for Hepatitis E virus, Encephalomyocarditis virus, *Francisella* or *Neospora caninum*, whereas two animals tested positive for *Toxoplasma gondii*. Moreover, a high prevalence of histological lesions has been found. Coypu can act as a host for several pathogens, including important agents for human and animal health, and surveillance is necessary to fully understand the biological role and the importance of coypu as a disease reservoir in our country.

**Abstract:**

*Myocastor coypus* is a pest animal present in Africa, Europe, North America and Asia that causes agricultural and ecological damages. Moreover, it has to be considered as a potential risk for public health. Forty-four coypus from the “Parco Naturale La Mandria” (Piedmont region, Northwest Italy) have been analysed. A complete necropsy and a whole histological evaluation of the liver, kidney and lung have been carried out on all the animals. Moreover, the positivity to Hepatitis E Virus (HEV), Encephalomyocarditis virus (EMCV), *Francisella* spp., *Toxoplasma gondii* and *Neospora caninum* have been investigated. None of the animal were positive for HEV, EMCV, *Francisella* spp. or *Neospora caninum*. Two animals tested positive for *Toxoplasma gondii*. A high presence of histological lesions has been identified in different organs, suggesting that lesions could be induced by different pathogens. As previously reported, coypu can act as a host for several pathogens, including important agents for human and animal health, and surveillance is necessary to fully understand the biological role and the importance of coypu as a disease reservoir in our country.

## 1. Introduction

The nutria, or coypu (*Myocastor coypus*), is a semiaquatic herbivorous rodent, originally native of South America, present currently in large feral populations in Africa, Europe, North America and Asia [1,2]. In Europe, the coypu was introduced principally for meat and fur production. In Italy the first introduction for fur farming dates back to 1928 by the National Institute of Rabbit husbandry in Alessandria (Piedmont, Northwest Italy) [3].

Coypu colonization of the natural environment is primarily due to intentional release actions and, minimally, to animals escaped from fur farms. During the following decades, coypu population, density and distribution have dramatically increased and, thanks to its ecological plasticity, even suboptimal habitats were colonized [4]. Due to its negative ecological impact, the coypu is currently considered a pest in the areas of introduction [5]. This animal usually lives next to water courses, in wetlands, riparian zones and coastlands [6]. Thereby, its burrowing behaviour undermines the banks of rivers, canals and dykes, whereas its feeding activity reduces plants’ biodiversity and cover, altering the water’s flowing speed and increasing the erosion of the banks [7]. 

Moreover, the damage to the habitat can negatively affect the reproduction of fishes, birds and invertebrates [8]. Coypu can also cause agricultural damages by feeding on crops, resulting in a great economic impact [9,10]. Due to the typical aspect of coypu ecology, this species is subjected to population control programs. Nevertheless, the population is still expanding [4,11].

Furthermore, the coypu can be infected by several pathogens and parasites. Some of them can be transmissible to humans and other animals. Several investigations focused on agents that might cause epizootics in wild populations and livestock, as well as in humans [5,12,13].

In this context, this research aims to evaluate the health status of the coypu in Northwest Italy and the presence of pathologies through gross necropsy examinations, histological and biomolecular investigations. The study focuses on viral, bacterial and parasitological pathogens, to investigate potential public health risks linked to coypu diffusion and to improve our knowledge about diseases affecting this species in the analysed area.

## 2. Materials and Methods

### 2.1. Animals and Samples Collection

Following the adoption of a regional animal containment programme (according to the D.G.R no. 74–6702 [08/03/2007] and subsequent amendments), 44 coypus from the “Parco Naturale La Mandria” (Piedmont region, Northwest Italy, 45°8′7″ N, 7°37′31″ E) were trapped with baited cage traps or shot. Trapped animals were euthanized with CO_2_, according to Italian National Bioethics Committee guidelines and to law no. 157 (02/11/1992) and subsequent amendments. The sample included 25 males and 19 females. For each animal, biometric features as weight (kg) and foot length (cm) were collected. All animals were subjected to a standard necropsy procedure. 

Samples of liver, kidney, lung and both eyeballs were fixed in 10% phosphate-buffered formalin for histological analysis and age determination.

Based on previous study in the same area [12], samples of liver, lung, heart and central nervous system (CNS) were collected and frozen at −20 °C for virological, bacteriological and parasitological investigations.

### 2.2. Age Estimation

Due to the impossibility of distinguishing between juveniles, adults and elderly by aspect, morphometric measures or dentition, the age was determined in 33 out of the 44 animals by dry eye-lens weight, according to the protocol proposed by Gosling and colleagues. In detail, the eyes were removed as soon after death as possible and placed in 10% buffered formalin. Eyes were kept in formalin for at least 1 month to allow the lens to harden. The lenses were extracted, and extraneous tissue was removed by rolling on absorbent paper and careful scraping with a blunt seeker. Any damaged lenses were discarded. Pairs of lenses were placed in a small crucible and dried in a forced draught at 80 °C for 22 h. Lenses were weighed on an analytical balance to an accuracy of 0.1 mg. The equation used to determine the age in months is log10 (age + 4.34 months) = 0.511 + 0.013 (mean lens weight) [14]. Based on the study of Pagnoni and Santolini [15], the animals were divided into two groups: juveniles (<8 months) and adults (8–12 months). 

### 2.3. Histological Analysis

The fixed samples were routinely processed and paraffin embedded. Three μm sections were stained with haematoxylin and eosin (HE) and with von Kossa staining, if calcium salts precipitations in the tissue were suspected.

### 2.4. Virological Analysis

The total RNA was extracted from liver and heart samples using TRIzol^®^ Reagent (Invitrogen™, ThermoFisher Scientific, Waltham, MA, USA), according to the manufacturer’s instructions. RNA extracted from liver was tested for the presence of Hepatitis Virus E (HEV) according to the protocol of Jothikumar [16]. Samples were considered negative for Ct (cycle threshold) values >38, doubtful if Ct was comprised between 36 and 38 and positive for Ct < 36. RNA extracted from heart was retrotranscribed through the High-Capacity cDNA Reverse Transcription Kit (Applied Biosystems™, ThermoFisher Scientific), and the obtained cDNAs were tested for the presence of the encephalomyocarditis virus (EMCV) using the protocol by Vanderhallen [17].

### 2.5. Bacteriological Analysis

The genomic DNA was extracted from liver samples using the QIAamp DNA Mini Kit (Qiagen, Hilden, Germany) according to the manufacturer’s protocol. The extracted templates were amplified using a primer set specific for *Francisella* spp. [18]. Specimens collected from 39 lungs were plated onto 5% sheep Blood Agar and Gassner Agar and incubated for 24 h at 37 °C. Bacterial isolates were identified to species level by means of an automated system using Vitek^®^ 2 Compact (bioMérieux, Inc., Durham, NC, USA), a testing system that combines an automated platform with an expansive database of clinically significant organisms.

### 2.6. Parasitological Analysis

Total genomic DNA was extracted from 25 mg of CNS homogenate, using the commercial kit NucleoSpin^®^ Tissue (Macherey-Nagel, Düren, Germany). The extracted templates were tested for *Toxoplasma gondii* [19] and *Neospora caninum* [20]. PCR was performed as previously reported [21].

### 2.7. Statistical Analysis

Statistical analyses were performed using GraphPad Prism (version 10.0.1, GraphPad Software, La Jolla, CA, USA). Fisher’s exact test was performed to determine non-random associations between sex and the recurrent pathologies detected in lung and kidney. A *p* value < 0.05 was considered statistically significant.

## 3. Results

### 3.1. Age Estimation

Out of 33 coypus considered for age determination, 32 were juveniles and only 1 was an adult (Table 1).

### 3.2. Histological Analysis

Out of 44 examined livers, 25 (56.8%) had no detectable microscopic lesions, whereas 18 (40.9%) showed one or more concurrent microscopic lesions (Table 2, Figure 1 and Figure 2). One sample (2.3%) was autolytic and impossible to evaluate.

The evaluation of the kidneys showed 21 (47.7%) samples without lesions and 23 (52.3%) affected from one or more detectable microscopic lesions (Table 3, Figure 3). No statistically significant association between lymphocytic interstitial nephritis and sex or age was detected.

Each of the 44 examined lungs showed the presence of concomitant different lesions (Table 4, Figure 4). No statistically significant association was found between perivascular lymphocytic infiltrate or BALT activation and sex or age.

### 3.3. Bacteriological, Virological and Parasitological Analyses

#### None of the Samples Tested Positive for HEV or EMCV

None of the tested animals were positive for *Francisella* spp., while 25 out of 39 analysed lungs (64.1%) were positive in the bacteriological analysis. Six of them (15.4%) revealed polymicrobial infection. The bacterial species identification of the other 19 positive samples is listed in Table 5. Due to the conservation methods of the lung samples, it has to be considered that an underestimation of bacterial identification is possible.

Two out of the thirty-five (5.7%) analysed animals were positive for *T. gondii*, whereas *N. caninum* infection has never been detected.

## 4. Discussion

All of the considered lungs had microscopical lesions. However, out of the 44 samples, only in 19 samples was it possible to isolate bacterial pathogens. Fifteen out of the nineteen (89.5%) analysed lungs showed the presence of lymphocytic infiltrates. Nine of them also showed the presence of lymphocytic infiltrate in the kidney and five of them in the liver. The other three individuals showed inflammatory lesions only in the liver.

As previously reported by Bollo and colleagues [12], cold and high humidity could be a predisposing factor for the high prevalence of pneumonia in wild coypu.

Histologically, kidneys mostly showed interstitial lymphocytic infiltrate (45.45%). This prevalence is higher than the one reported in another study conducted in the same region (10.1%) [12], and it could be caused by infection or immune-mediated diseases, such as Leptospirosis, as previously reported in other populations of coypu serologically investigated [12,22,23,24].

Livers showed microscopical lesions, with inflammatory infiltrations, mostly lymphocytic. The aetiology of those lesions can be related to inflammatory and degenerative processes, differently from the results obtained by Bollo and colleagues [12].

The investigation for EMCV’s RNA was negative. In previous works, seropositivity against EMCV was found in Argentina [13] and Italy [12], whereas it was not detected in the USA [25].

PCR for *Francisella* spp. gave negative results. Similar results have been serologically obtained in two different studies made in Louisiana (USA) and Argentina, in which no antibodies against *F. tularensis* were found [13,25].

All the samples analysed for HEV and *Neospora caninum* were negative and, to the authors’ best knowledge, positivity in this species has never been demonstrated.

Worthy of interest, two animals were positive for *Toxoplasma gondii*, which can infect all warm-blooded animals and is the aetiological agent of a major zoonosis. In the present research, one of the two animals found to be positive for *T. gondii* showed an interstitial lymphocytic nephritis and a lymphocytic perivascular and parenchymal infiltrate in the liver, even if, normally, animals challenged with *Toxoplasma* do not develop referable clinical signs [5,12]. The other positive animal showed no microscopical findings. The prevalence of *Toxoplasma* in the examined area (4.5%) is very low compared to Nardoni and colleagues’ study (59.4%) [5]. A coypu infected with *Toxoplasma gondii* is a potential contagion source for other scavengers and can be used in the analysed area for monitoring the quantity of oocysts [5]. Human infection during pregnancy may be extremely dangerous for the foetus, and this parasite in immunocompromised patients can cause a life-threatening encephalitis [26]. New findings suggest that *Toxoplasma* can actually cause changes in memory, learning, behaviour and anxiety [27].

Currently, the coypu is mostly raised in South America for fur production. Meat consumption of the coypu is considered a by-product, but there are new studies that are considering coypu meat as a new novel and exotic food [28]. In this scenario, the importance of coypu infections with *Toxoplasma* and other pathogens such as HEV acquires even more relevance.

## 5. Conclusions

In conclusion, the coypu is an acknowledged threat for both the environment and animals. It can act as a host for several pathogens, including important agents for human and animal health [5,9,12]. Therefore, further investigation for viral, bacterial and parasitic surveillance are necessary, to fully understand the biological role and the importance of coypu as a disease reservoir in our country.

## Figures and Tables

**Figure 1 animals-14-00245-f001:**
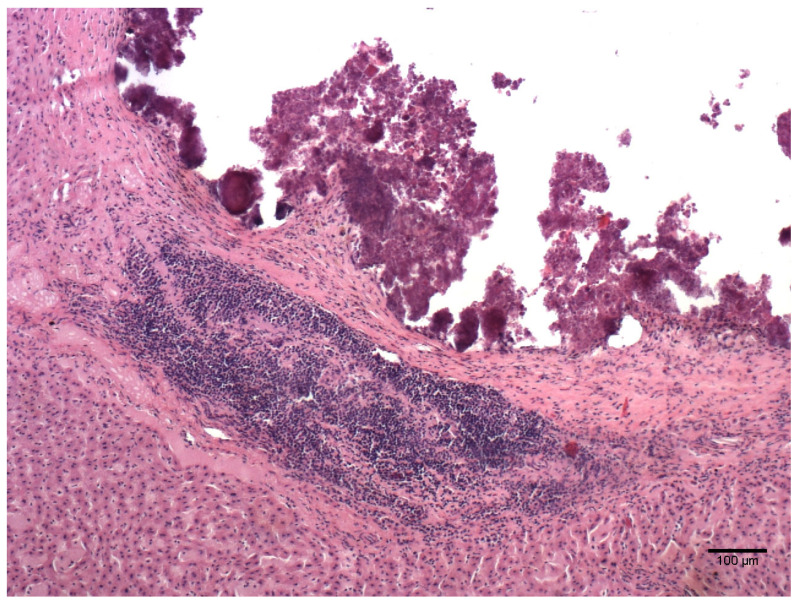
*Myocastor coypus* liver: granuloma with necrotic area surrounded by connective tissue and lymphocytic infiltrate (HE).

**Figure 2 animals-14-00245-f002:**
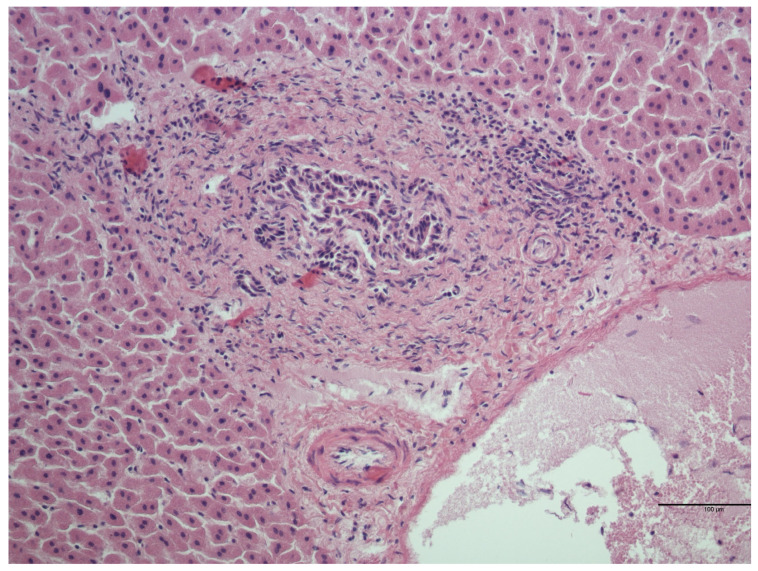
*Myocastor coypus* liver: mild activation of the periportal lymphoid tissue (HE).

**Figure 3 animals-14-00245-f003:**
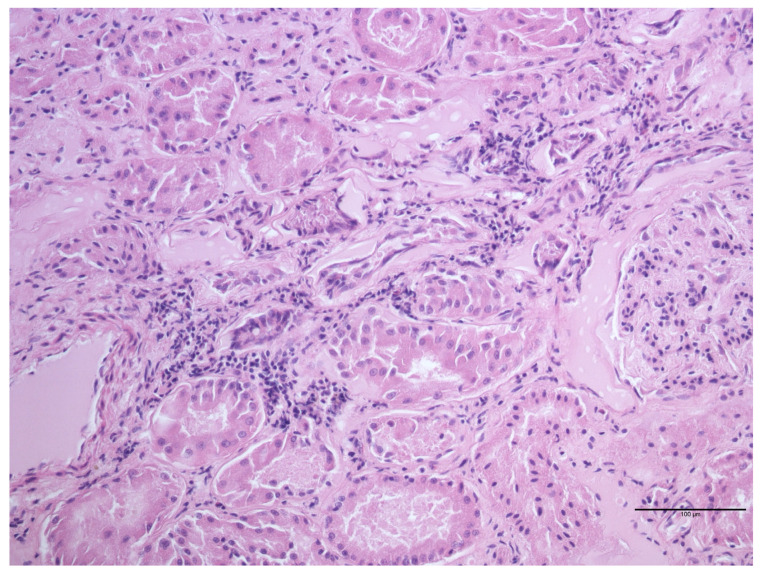
*Myocastor coypus* kidney: mild interstitial lymphoid nephritis (HE).

**Figure 4 animals-14-00245-f004:**
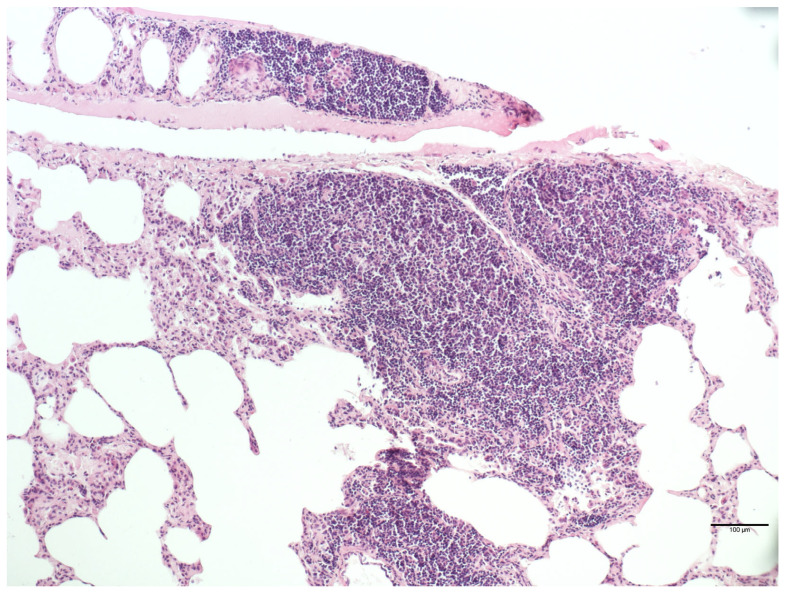
*Myocastor coypus* lung: severe parenchymal lymphocytic infiltrate (HE).

**Table 1 animals-14-00245-t001:** Mean weight of the dry eye lens (mg), standard deviation, age in months and age class in tested coypus. NA: not applicable.

Coypus ID	Mean Weight (mg)	Standard Deviation (mg)	Age (Months)	Age Class
1	NA	NA	NA	NA
2	NA	NA	NA	NA
3	NA	NA	NA	NA
4	NA	NA	NA	NA
5	NA	NA	NA	NA
6	11.5	4.95	<1	Juvenile
7	7.5	4.95	<1	Juvenile
8	6	1.41	<1	Juvenile
9	7	1.41	<1	Juvenile
10	5.5	0.71	<1	Juvenile
11	NA	NA	NA	NA
12	NA	NA	NA	NA
13	35.5	9.19	5.0	Juvenile
14	34.5	2.12	4.8	Juvenile
15	46	NA	8.5	Adult
16	44	NA	7.8	Juvenile
17	36	NA	5.2	Juvenile
18	27.5	4.95	3.0	Juvenile
19	21	2.83	1.7	Juvenile
20	38	5.66	5.8	Juvenile
21	44	7.07	7.8	Juvenile
22	24.5	0.71	2.4	Juvenile
23	6	2.83	<1	Juvenile
24	13.5	2.12	0.5	Juvenile
25	13.5	4.95	0.5	Juvenile
26	13	1.41	0.4	Juvenile
27	6	4.24	<1	Juvenile
28	12.5	2.12	0.4	Juvenile
29	15.5	2.12	0.8	Juvenile
30	10.5	0.71	0.1	Juvenile
31	9.5	4.95	0.0	Juvenile
32	5	4.24	<1	Juvenile
33	20	5.66	1.6	Juvenile
34	10	0.00	0.0	Juvenile
35	21	2.83	1.7	Juvenile
36	28	2.83	3.2	Juvenile
37	7.5	0.71	<1	Juvenile
38	14	0.00	0.6	Juvenile
39	NA	NA	NA	NA
40	24.5	0.71	2.4	Juvenile
41	NA	NA	NA	NA
42	13.5	2.12	0.5	Juvenile
43	NA	NA	NA	NA
44	NA	NA	NA	NA

**Table 2 animals-14-00245-t002:** Frequency and characteristics of the histological lesions detected in liver samples.

Lesion	Number of Samples Positive for the Lesion/Total Number of Microscopical Lesions (%)
Periportal lymphoid tissue activation	8/18 (44.4%)
Parenchymal lymphocytic infiltrate	7/18 (38.9%)
Perivascular lymphocytic infiltrate	3/18 (16.7%)
Macrophage infiltration	1/18 (5.6%)
Multifocal granuloma	1/18 (5.6%)

**Table 3 animals-14-00245-t003:** Frequency and characteristics of the histological lesions detected in kidney samples.

Lesion	Number of Samples Positive for the Lesion/Total Number of Microscopical Lesions (%)
Interstitial lymphocytic nephritis	20/23 (87.0%)
Urine crystals	2/23 (8.7%)
Perivascular lymphocytic infiltrate	1/23 (4.3%)
Cyst with focal lymphocytic infiltrate	1/23 (4.3%)
Interstitial lymphocytic and eosinophilic nephritis	1/23 (4.3%)
Lymphocytic infiltrate into perirenal fat	1/23 (4.3%)

**Table 4 animals-14-00245-t004:** Frequency and characteristics of the histological lesions detected in lung samples.

Lesion	Number of Samples Positive for the Lesion/Total Number of Microscopical Lesions (%)
Emphysema	44/44 (100%)
Oedema	36/44 (81.8%)
Parenchymal lymphocytic infiltrate	36/44 (81.8%)
Perivascular lymphocytic infiltrate	32/44 (72.7%)
BALT activation	27/44 (61.4%)
Alveolar thickening	12/44 (27.3%)
Atelectasis	9/44 (20.5%)
Parenchymal lymphocytic and eosinophilic infiltrate	2/44 (4.5%)
Lymphocytic bronchitis	2/44 (4.5%)
Parenchymal neutrophilic infiltrate	1/44 (2.3%)
Focal haemorrhages	1/44 (2.3%)

**Table 5 animals-14-00245-t005:** Isolated bacteria.

Isolated Bacteria	Number of Samples Positive/Total Number of Analysed Lungs (%)
*Enterococcus* spp.	4/19 (21.0%)
*Enterococcus hirae*	2/19 (10.5%)
*Pseudomonas fluorescens*	2/19 (10.5%)
*Nocardia* spp.	2/19 (10.5%)
*Enterococcus durans*	1/19 (5.3%)
*Pseudomonas mendocina*	1/19 (5.3%)
*Achromobacter xylosoxidans*	1/19 (5.3%)
*Brevibacillus laterosporus*	1/19 (5.3%)
*Corynebacterium propinquum*	1/19 (5.3%)
*Corynebacterium pseudodiphthericum*	1/19 (5.3%)
*Ochrobactrum anthropi*	1/19 (5.3%)
*Streptococcus aginosus*	1/19 (5.3%)
Non-identifiable	1/19 (5.3%)

## Data Availability

Data are contained within the article.

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
