# Peer review of "A Health Status Update of Myocastor coypus in Northern Italy"

_animals, 2024, doi:10.3390/ani14020245_

Round 1

Reviewer 1 Report

Comments and Suggestions for Authors

Dear authors, this is an interesting and important research, summarizing pathological findings in Italian nutria, and therefore contributes to actual research questions in the topic of invasive alien species. Apart from histological diagnosis of liver, lung and kidneys in each animal, RNA-based testing for HEV from liver tissue and cDNA-testing for EMCV was done from heart tissue. Bacteriological investigation comprised testing for Francisella (PCR) from liver tissue samples and Agar-incubation for lung tissue samples. Parasitological investigation was done with CNS homogenate for Toxoplasma gondii and Neospora caninum (PCR). Overall, there were only two positive findings for Toxoplasma gondii.

One has to consider that the number of 44 animals is low, and the number of bacteriological/parasitological/virological tested animals is even lower and should be clearly stated for all methods used. Being so, it is questionable if the study design is good enough to detect the pathogens that authors were aiming at. Here, a clear strategy, e.g. hypothesis and prevalence of named pathogens in the wild, is missing. The isolation of bacteria of formerly frozen lung tissue samples might lead to misinterpretation, and this should be discussed, as well as the outcomes and potential causes for pneumonia that was detected.

There is no indication of how the pathogens tested for were chosen, and especially since the importance nutria as an indicator species for Echinococcus spp. is mentioned (L56), it is questionable why the authors do not address this topic. Here, more information should be presented on the hypotheses that the authors assumed and the decision-making process (beginning within the background in the part of introduction, and referring to in the methods section).

For age estimation, detailed information on weight lense and age would be necessary, e.g. in a table, since the weight lense method is more reliable in young animals. It would be helpful to have standard deviations shown. Definitely, more information is necessary to strengthen the papers content, since little is known of the suitability of Goslings method in further studies.

Also, it weakens the paper that underlying causes of the histological findings in kidneys could not be further elaborated. Pictures of the main histological findings would be helpful.

For publication, hypotheses and follow-up research should be elaborated in more detail. Maybe it should be considered to publish not as an Original Article, but as a Short Communication, to respect the limitation of results obtained.

Author Response

One has to consider that the number of 44 animals is low, and the number of bacteriological/parasitological/virological tested animals is even lower and should be clearly stated for all methods used. Being so, it is questionable if the study design is good enough to detect the pathogens that authors were aiming at. Here, a clear strategy, e.g. hypothesis and prevalence of named pathogens in the wild, is missing.

The ”la Mandria Park” is an Italian national park. The number of animals subjected to the regional animal containment programme is based on the estimated population in that area. No more animals were available in that area. The number of samples tested for each method is clearly stated in the results.

The isolation of bacteria of formerly frozen lung tissue samples might lead to misinterpretation, and this should be discussed, as well as the outcomes and potential causes for pneumonia that was detected.

A sentence underlines the possibility to have underestimate the presence of bacteria in the lung due to the conservation method of sample has been added, as suggested.

There is no indication of how the pathogens tested for were chosen, and especially since the importance nutria as an indicator species for Echinococcus spp. is mentioned (L56), it is questionable why the authors do not address this topic. Here, more information should be presented on the hypotheses that the authors assumed and the decision-making process (beginning within the background in the part of introduction, and referring to in the methods section).

L59 We selected the pathogens to test referring to previous study in the same area. To our best knowledge, Echinococcus spp. has been found in coypu in central Europe, in Italy there are not published positivity for this parasite in coypu and no lesions due to echinococcus have been found during necropsy.

For age estimation, detailed information on weight lense and age would be necessary, e.g. in a table, since the weight lense method is more reliable in young animals. It would be helpful to have standard deviations shown. Definitely, more information is necessary to strengthen the papers content, since little is known of the suitability of Goslings method in further studies.

L61-70 and table 1: detailed informations about the used methods have been added. Moreover, a table with mean weight of the dry eye lens (mg), standard deviation (mg), age in months and age class in tested coypus has been added. The age classification has revealed that almost all the animals were young.

Also, it weakens the paper that underlying causes of the histological findings in kidneys could not be further elaborated. Pictures of the main histological findings would be helpful.

Pictures of the main histological findings have been added, as suggested.

Reviewer 2 Report

Comments and Suggestions for Authors

Contents of the MS are useful, but the MS requires a deep revision. Particularly, an English revision is strongly needed by a native speaker.

11)      The title could be changed into: A health status update of Myocastor coypus in Norther Italy.

22)      Use the term “coypu” instead of “Nutria” throughout the text

33)      Line 15. “NNutria”

44)      Line 34. Change “nowadays” with “currently”.

55)      Line 35. The reference is very old. Please, check: Schertler, A., Rabitsch, W., Moser, D., Wessely, J., & Essl, F. (2020). The potential current distribution of the coypu (Myocastor coypus) in Europe and climate change induced shifts in the near future. NeoBiota58, 129-160; Pedruzzi, L., Schertler, A., Giuntini, S., Leggiero, I., & Mori, E. (2022). An update on the distribution of the coypu, Myocastor coypus, in Asia and Africa through published literature, citizen-science and online platforms. Mammalian Biology102(1), 109-118.

66)      Line 37. Clarify where is Alessandria, and please update the reference. Alessandria is not in Latium.

77)      Lines 41-42: please clarify the impacts of alien coypus.

88)      Line 43. Why do you say “usually”?

99)      Line 46. Add a reference.

110)   Line 51. You used “still”, but you referred to a paper published in 2007.

111)   Lines 81-82. Is there any way to distinguish between adults, subadults and juveniles only by looking at the individuals?

112)    The reference list needs a severe revision, most species name are written not in italics, and with the species epithet with capital letters. Some journal names are shortened, some other not.

Comments on the Quality of English Language

English needs some deep revisions.

Author Response

11)      The title could be changed into: A health status update of Myocastor coypus in Norther Italy.

The title has been changed, as suggested.

22)      Use the term “coypu” instead of “Nutria” throughout the text

“Nutria” has been changed in “coypu” throughout the text, as suggested.

33)      Line 15. “NNutria”

The typo has been corrected, as suggested.

44)      Line 34. Change “nowadays” with “currently”.

“nowadays” has been changes with “currently”, as suggested.

55)      Line 35. The reference is very old. Please, check: Schertler, A., Rabitsch, W., Moser, D., Wessely, J., & Essl, F. (2020). The potential current distribution of the coypu (Myocastor coypus) in Europe and climate change induced shifts in the near future. NeoBiota58, 129-160; Pedruzzi, L., Schertler, A., Giuntini, S., Leggiero, I., & Mori, E. (2022). An update on the distribution of the coypu, Myocastor coypus, in Asia and Africa through published literature, citizen-science and online platforms. Mammalian Biology102(1), 109-118.

The reference of Pedruzi and colleagues has been added, as suggested.

66)      Line 37. Clarify where is Alessandria, and please update the reference. Alessandria is not in Latium.

Alessandria is in Piedmont. The reference is referred to Latium but, in the introduction, is described the introduction of Nutria in Italy. No previous indexed references are available for this topic.

77)      Lines 41-42: please clarify the impacts of alien coypus.

88)      Line 43. Why do you say “usually”?

The word has been deleted

99)      Line 46. Add a reference.

110)   Line 51. You used “still”, but you referred to a paper published in 2007.

A recent reference of the Piedmont region with the trend population of Nutria has been added.

111)   Lines 81-82. Is there any way to distinguish between adults, subadults and juveniles only by looking at the individuals?

The sentence: “Due to the impossibility to distinguish between juveniles, adults and elderly by aspect, morphometric measures or dentition, the age was determined in 33 out of the 44 animals by dry eye lens weight, according to the protocol proposed by Gosling and colleagues” has been edded to better clirify why is necessary to use the dry eye lens method.

Round 2

Reviewer 1 Report

Comments and Suggestions for Authors

Dear authors, the additional information has improved the paper, thank you for your work.

Nevertheless, I believe that the sampling strategy should contain more animals, to get numbers of samples that come near to incidences. Either additional surrounding areas or timeframe would be appropriate here.

I wish the authors all the best.

Reviewer 2 Report

Comments and Suggestions for Authors

Authors have addressed all of my previous comments. Thus, their MS can be accepted for publication.